# Actin-based protrusions of migrating neutrophils are intrinsically lamellar and facilitate direction changes

Lillian K Fritz-Laylin[1†‡], Megan Riel-Mehan[2†], Bi-Chang Chen[3§], Samuel J Lord[1], Thomas D Goddard[4], Thomas E Ferrin[4], Susan M Nicholson-Dykstra[5#], Henry Higgs[5], Graham T Johnson[2,6], Eric Betzig[3], R Dyche Mullins[1*]

[1]Department of Cellular and Molecular Pharmacology, Howard Hughes Medical Institute, University of California, San Francisco, San Francisco, United States; [2]Department of Bioengineering and Therapeutic Sciences, University of California, San Francisco, San Francisco, United States; [3]Janelia Research Campus, Howard Hughes Medical Institute, Ashburn, United States; [4]Department of Pharmaceutical Chemistry, University of California, San Francisco, San Francisco, United States; [5]Department of Biochemistry and Cell Biology, Dartmouth Geisel School of Medicine, Hanover, United States; [6]Animated Cell, Allen Institute for Cell Science, Seattle, United States

**\*For correspondence:**
Dyche.Mullins@ucsf.edu

†These authors contributed equally to this work

**Present address:** ‡Department of Biology, University of Massachusetts, Amherst, United States; §Research Center for Applied Sciences, Academia Sinica, Taipei, Taiwan; #Thornton High School, Thornton, United States

**Competing interests:** The authors declare that no competing interests exist.

**Abstract** Leukocytes and other amoeboid cells change shape as they move, forming highly dynamic, actin-filled pseudopods. Although we understand much about the architecture and dynamics of thin lamellipodia made by slow-moving cells on flat surfaces, conventional light microscopy lacks the spatial and temporal resolution required to track complex pseudopods of cells moving in three dimensions. We therefore employed lattice light sheet microscopy to perform three-dimensional, time-lapse imaging of neutrophil-like HL-60 cells crawling through collagen matrices. To analyze three-dimensional pseudopods we: (i) developed fluorescent probe combinations that distinguish cortical actin from dynamic, pseudopod-forming actin networks, and (ii) adapted molecular visualization tools from structural biology to render and analyze complex cell surfaces. Surprisingly, three-dimensional pseudopods turn out to be composed of thin (<0.75 μm), flat sheets that sometimes interleave to form rosettes. Their laminar nature is not templated by an external surface, but likely reflects a linear arrangement of regulatory molecules. Although we find that Arp2/3-dependent pseudopods are dispensable for three-dimensional locomotion, their elimination dramatically decreases the frequency of cell turning, and pseudopod dynamics increase when cells change direction, highlighting the important role pseudopods play in pathfinding.
DOI: https://doi.org/10.7554/eLife.26990.001

## Introduction

Three hundred years, ago Anton van Leeuwenhoek's simple microscope revealed 'animacules' changing shape as they moved through a drop of water (*van Leewenhoeck, 1677*). Subsequent breakthroughs in microscope technology and molecular biology have strengthened the link between cell morphology and locomotion, and we now know that crawling cells from many eukaryotic phyla create a variety of dynamic protrusions that project forward, in the direction of migration.

Thin, flat protrusions produced by strongly adherent cells are generally called lamellipodia. Previous work, mainly on fibroblasts and epithelial cells, has established that movement of leading-edge lamellipodia is driven by growth of branched actin networks nucleated and crosslinked by the Arp2/

3 complex. As they advance, these lamellipodia promote creation of new focal adhesions that provide traction and support continued movement (*Case and Waterman, 2015*). This mode of cell migration appears to be restricted to the animal lineage, where it relies on the presence of specific ligands in the extracellular environment and plays essential roles in embryonic development, wound healing, and tissue homeostasis.

A more widely dispersed form of cell crawling—probably present in the last common ancestor of all eukaryotes (*Fritz-Laylin et al., 2017*)—requires only weak, non-specific interaction with the extracellular environment and can achieve speeds several orders of magnitude faster than adhesion-based movement (*Loomis et al., 2012*; *Preston and King, 1978*; *Lämmermann et al., 2008*; *Buenemann et al., 2010*; *Butler et al., 2010*; *Loomis et al., 2012*; *Petrie and Yamada, 2015*; *Fritz-Laylin et al., 2017*). This mode of cell migration is associated with formation of complex, three-dimensional pseudopods filled with branched networks of actin filaments, nucleated and organized by the Arp2/3 complex. These three-dimensional pseudopods may contribute to forward motion of crawling and swimming cells (*Van Haastert, 2011*), but recent work has revealed that disrupting pseudopod formation can *increase* the speed and persistence of migration (*Leithner et al., 2016*; *Vargas et al., 2016*), implying that complex pseudopods might be more important for exploring the external environment than for driving forward motion.

We know less about the morphology and function of complex pseudopods than adherent lamellipodia in part due to the technical limitations of live-cell light microscopy. Confocal and total internal reflection fluorescence (TIRF) microscopy, for example, provide adequate views of adherent lamellipodia because these structures are: (i) slow-moving, (ii) thin, and (iii) closely adhered to the microscope coverslip. In contrast, the complex pseudopods created by fast-moving cells: (i) grow quickly (10–100 um/min), (ii) adopt complex three-dimensional shapes, and (iii) often spend their lives far from a coverslip surface. In addition, photobleaching must be minimized in order to observe living cells for sufficient time to track their complex three-dimensional migration.

Although some work has suggested that two-dimensional lamellipodia of adherent cells reflect the flatness of the surface to which they are attached (*Burnette et al., 2014*), no mechanisms have been proposed to explain the more complex morphologies of three-dimensional pseudopods created by fast, weakly adherent cells—especially when crawling through irregular environments. Moreover, it remains unclear whether these complex pseudopods are completely amorphous, or whether they share common, underlying structural features that might shed light on their functions and mechanisms of assembly.

To address these issues, we employed the recently developed lattice light sheet microscope (*Chen et al., 2014*), which offers a unique combination of three-dimensional imaging with high spatial and temporal resolution and low phototoxicity. To render and interpret the large data sets produced by lattice light sheet microscopy, we developed new visualization software and combined it with biochemically defined molecular probes to detect and characterize pseudopods in living cells.

Using these new tools, we discovered that pseudopods created by fast-moving neutrophils are not entirely amorphous, but represent various arrangements of a single structural motif: a free-standing lamellar sheet. Thus, complex, three-dimensional pseudopods that appear amorphous in widefield and confocal microscopy turn out to be rosettes formed of multiple, interleaved lamella. Unlike adherent lamellipodia, these free-standing lamellar building blocks are not templated from a flat surface and their morphology strongly suggests that they arise from a linear arrangement of regulatory molecules associated with the plasma membrane. In addition, our automated methods for detecting and quantifying the size of complex pseudopods reveal that these dynamic cellular structures play a key role in cellular pathfinding.

## Results

### Surface rendering of lattice light sheet data reveals three-dimensional dynamics of membranes and actin networks in migrating neutrophils

Neutrophil-like HL-60 cells build pseudopods that are filled with polymerized actin, easily visualized in fixed cells using phalloidin (*Figure 2—figure supplement 1*, *Figure 3—figure supplement 1*). To visualize pseudopod dynamics in actively migrating cells, we constructed HL-60 cell lines that stably express fluorescent markers for actin filaments and the plasma membrane (*Figure 1*). No actin probe

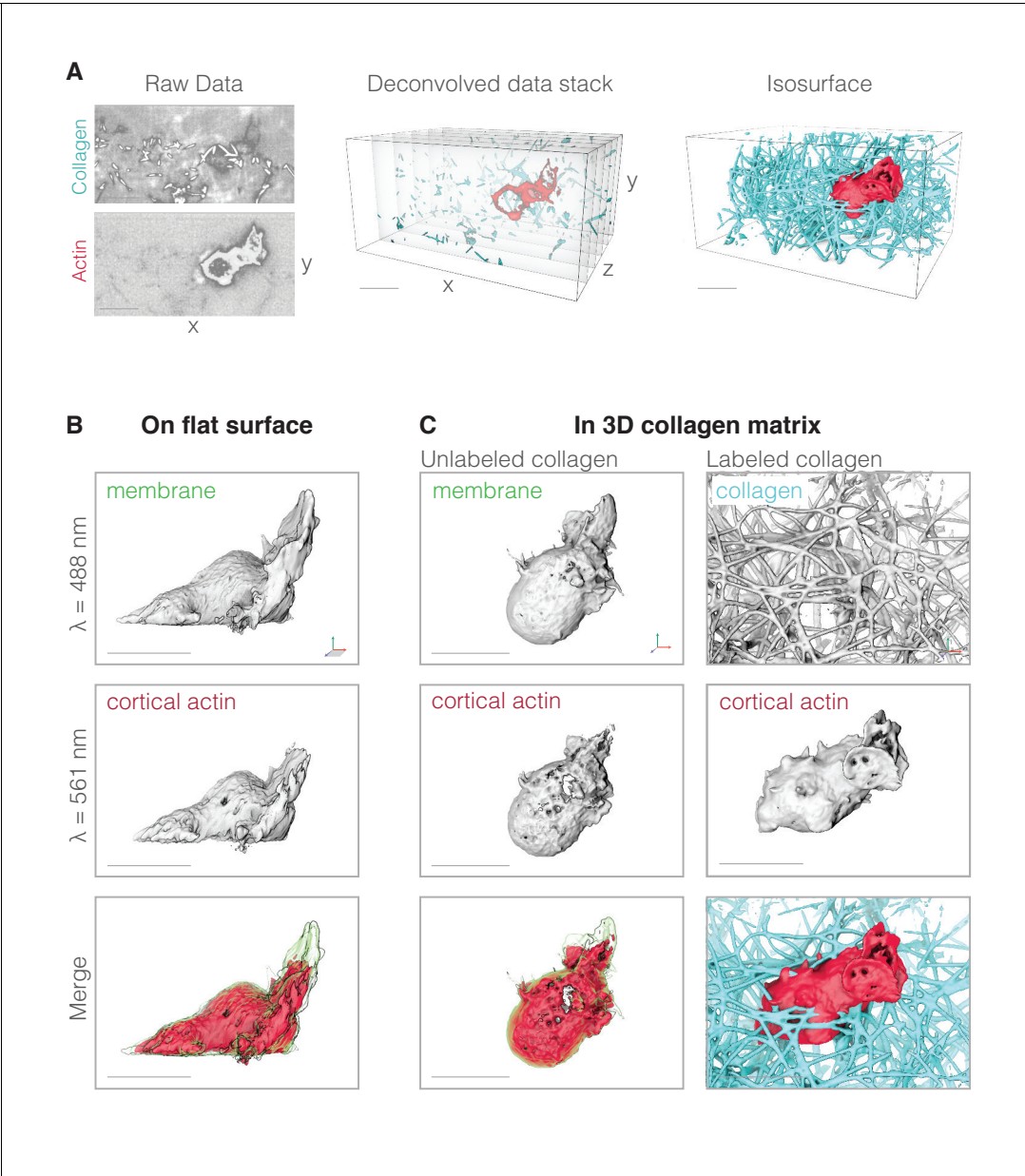

**Figure 1.** Data processing and visualization. (A) Data processing pipeline. Left: representative example of raw LLSm data showing a single image plane that passes through a neutrophil-like HL-60 cell (bottom panel) migrating through a collagen network (top panel). Resolution is ~230 in X and Y,~370 in Z. Collagen is directly labeled with fluorescein and the cell's actin cytoskeleton is highlighted using a utrophin-based probe (see below). Middle: the raw data are deconvolved into a pair of single-color vertical image stacks (here shown merged). Right: the vertical image stacks are then rendered as three-dimensional isosurfaces in UCSF Chimera. (B) Representative fluorescence iso-surface images of an HL-60 cell expressing markers for the plasma membrane (top) and actin cytoskeleton (middle) and crawling across a fibronectin-coated glass coverslip. Top: fluorescence of plasma membrane-anchored palmitoylated m-Emerald. Middle: fluorescence of cortical actin filaments in the cell body labeled by Utrophin-mCherry. Bottom: false-color, three-dimensional overlay of the membrane-mEmerald and Utrophin-mCherry images. (C) Representative fluorescence iso-surface images of HL-60 cells migrating through unlabeled (left column) and fluorescently labeled (right column) three-dimensional collagen networks. Scale bars = 10 μm. Axes indicate relative orientation across rotated views. For cells crawling on a 2D surface, the orientation of the coverslip is indicated by gray shading. Unless otherwise specified, cells were illuminated with 488 and 560 nm light at 37C in 1 × HBSS supplemented with 3% FBS, 1 × pen/strep, and 40 nM of the tripeptide formyl-MLP (to stimulation migration).

DOI: https://doi.org/10.7554/eLife.26990.002

The following figure supplements are available for figure 1:

**Figure supplement 1.** A Lifeact-based fluorescent probe does not label filamentous actin in HL-60 pseudopods.

DOI: https://doi.org/10.7554/eLife.26990.003

*Figure 1 continued on next page*

*Figure 1 continued*

**Figure supplement 2.** Comparison of isosurface views normal to the XY, YZ, and XZ imaging planes, illustrating the near-isotropic resolution of lattice light sheet microscopy.
DOI: https://doi.org/10.7554/eLife.26990.004
**Figure supplement 3.** Mesh processing for figures, showing both the polygon counts (or 'three-dimensional Mesh') and the rendered images.
DOI: https://doi.org/10.7554/eLife.26990.005

binds exclusively to the dynamic, branched actin networks of the pseudopod, including the widely used probe, Lifeact (*Figure 1—figure supplement 1*). Therefore, we chose the utrophin-based actin probe, which binds preferentially to cortical actin surrounding the cell body and is largely excluded from the more dynamic, branched actin networks that drive pseudopod extension (*Belin et al., 2014*). When combined with a membrane marker, the selectivity of our utrophin-based actin probe enabled us to develop automated methods for identifying and tracking three-dimensional pseudopods in moving cells (see below). This 'negative space' approach to identifying pseudopods also avoids problems associated with the speed at which probe molecules diffuse through dense actin networks (*Belin et al., 2014*).

We used lattice light sheet microscopy to create time-lapse sequences of high-resolution (230 × 230×370 nm [*Chen et al., 2014*]), nearly isotropic, three-dimensional images of HL-60 cells crawling across fibronectin-coated coverslips or moving through random networks of collagen fibers (*Figure 1* and *Figure 1—figure supplement 2*). Handling the raw data, we quickly realized that the ability of lattice light sheet microscopy to provide new insights into cell biology is hampered, in part, by the lack of computational tools for visualizing and analyzing three-dimensional fluorescence intensity data. The most widely used tools for analyzing microscopy data can perform basic operations such as thresholding, normalization, alignment, and drift correction, but these tools were designed to handle two-dimensional images. Rather than attempting to extend the capabilities of a primarily two-dimensional computational platform, we instead adapted software designed from the outset to manipulate and analyze three-dimensional data sets. Specifically, we chose the molecular visualization program UCSF Chimera (*Pettersen et al., 2004*) for this purpose because it is freely available and it incorporates the collective experience of the structural biology community in rendering and manipulating three-dimensional data.

Briefly, we added to UCSF Chimera the capability to import a time series of three-dimensional fluorescence intensity datasets. Once imported, these data can be rendered and manipulated by native UCSF Chimera functions, including elements of the 'vseries' toolkit, which perform volumetric versions of thresholding, normalization, alignment, and fitting. In addition, the capacity of UCSF Chimera to render surfaces of equal fluorescence intensity dramatically enhances the ability to judge proximity and/or co-localization of fluorescent probes in three dimensions. Finally, inspired by rendering tools available in professional animation software (e.g. Cinema4D; www.maxon.net), we added a tonal shading technique, called ambient occlusion (*Tarini et al., 2006*), to a recent UCSF Chimera release (http://www.rbvi.ucsf.edu/chimera/data/ambient-jul2014/ambient.html). Tonal surface rendering with ambient occlusion makes cellular structures more easily interpretable by emphasizing the spatial relationships of three-dimensional surfaces and variations in surface texture. The enhanced ability to detect and interpret textures and protrusions can be seen by comparing traditional maximum intensity projection and rendered versions of confocal data (*Figure 3—figure supplement 1*) and similar renderings of the lattice light sheet data (*Figure 3—figure supplement 2*).

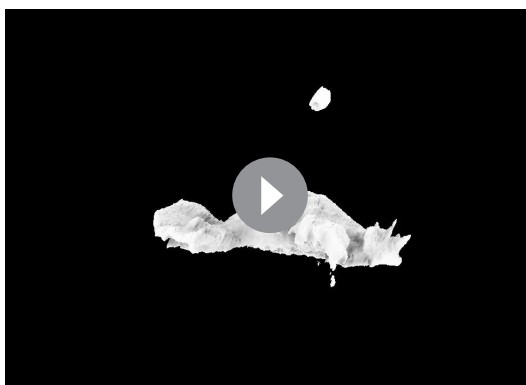

**Video 1.** Example of lamellar pseudopods formed by cells crawling on a flat surface (fibronectin-coated glass coverslip). Video plays at 10.5 × real time.
DOI: https://doi.org/10.7554/eLife.26990.006

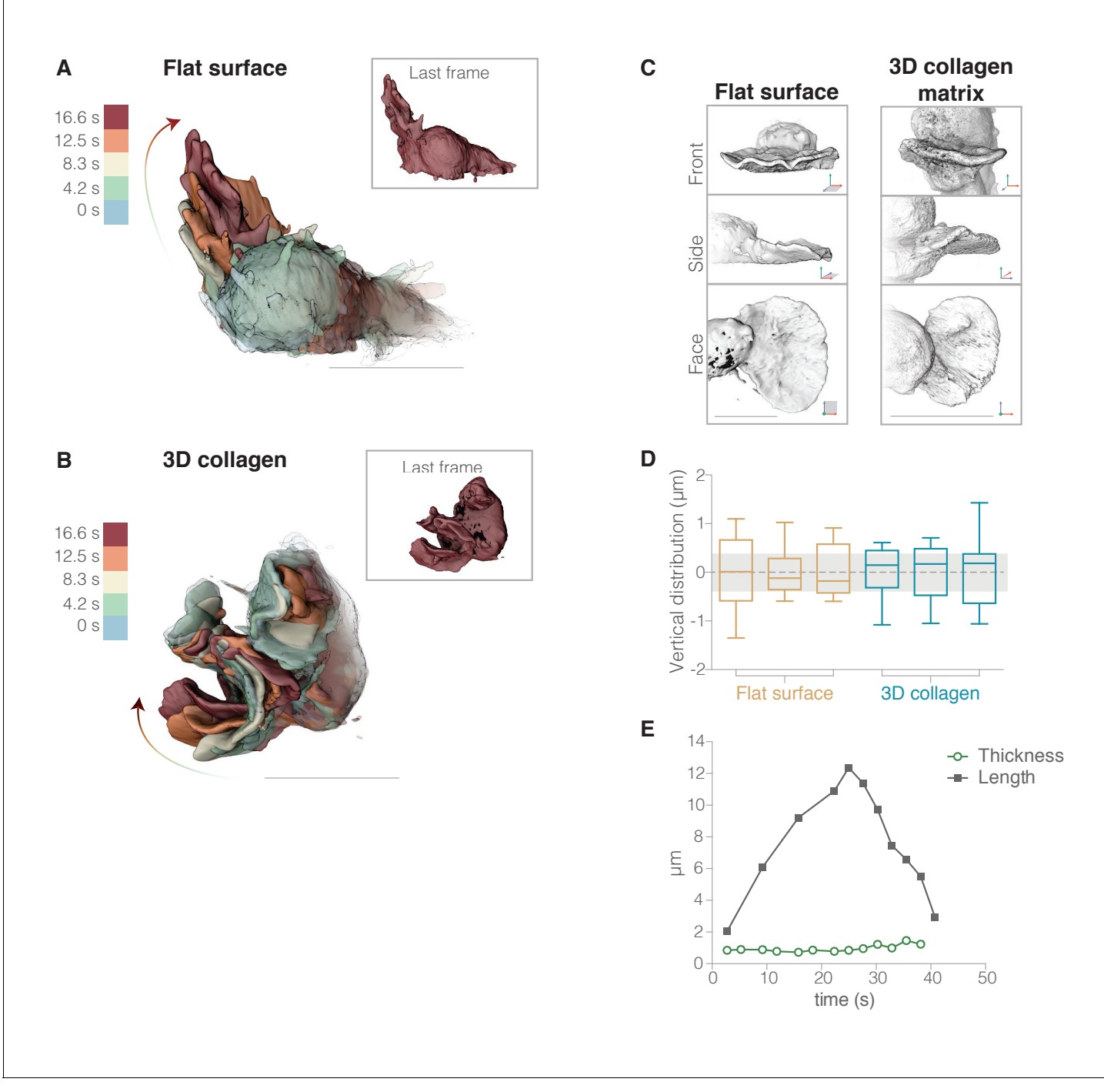

**Figure 2.** Neutrophils form lamellar pseudopods regardless of whether they are crawling on two-dimensional surfaces or moving through complex three-dimensional environments. (A–B) To illustrate the morphology of rapidly growing membrane protrusions we overlaid multiple fluorescence iso-surface images taken at different time points of individual HL-60 cells expressing a membrane probe (palmitoylated m-Emerald). Individual surface renderings are tinted according to time. Insets show each cell at the final time point. (A) HL-60 cell crawling across a fibronectin-coated glass coverslip. (B) HL-60 cell crawling through a polymerized collagen mesh. (C) Three-dimensional surface renderings of two pseudopods from two different cells, one crawling on fibronectin-coated glass (left) and one moving through a collagen fiber matrix (right). Each LLSm dataset is rendered from three different viewing angles: en face (top), from the side (middle), and from above (bottom). For more examples, see *Figure 2—figure supplement 1*. (D) Box and whisker plot (with median, quartiles, and range) of vertical error (estimate of flatness) for each pseudopod, calculated by measuring the average Euclidean distance between the original points on the leading edge and a reference plane with a width equal to the average measured width of lamellar pseudopods (gray band). The first three are measurements from cells crawling across a flat surface (brown), and the next three from cells crawling through a polymerized collagen network (blue). (E) Example of a lamellar pseudopod retaining a similar thickness (green circles) during
*Figure 2 continued on next page*

*Figure 2 continued*
extension and retraction as indicated by the distance from the tip of the pseudopod to the cell body (black squares). Axes as indicated in *Figure 1*. All scale bars = 10 μm.
DOI: https://doi.org/10.7554/eLife.26990.016
The following figure supplement is available for figure 2:
**Figure supplement 1.** Neutrophils form lamellar pseudopods.
DOI: https://doi.org/10.7554/eLife.26990.017

## Dynamic pseudopods are composed of intrinsically lamellar elements

Using these visualization tools, we observed two distinctive types of dynamic membrane protrusion at the leading edge of HL-60 cells crawling on fibronectin-coated coverslips: (i) thin, sheet-like lamina (*Figure 2* and *Videos 1–6*), and (ii) 'rosettes' formed by the coincidence of multiple lamina (*Figure 3*, *Videos 7–10*). The morphology of these protrusions differ from the spherical blebs produced when the plasma membrane detaches from the underlying cortex (*Charras and Paluch, 2008*), instead they resemble membrane movements driven by rapid assembly of filamentous actin. Using confocal microscopy of fixed cells labeled with phalloidin, we verified that leading-edge sheets and rosettes are filled with dense actin networks (*Figure 2—figure supplement 1* and *Figure 3—figure supplement 1*). Dynamic sheets and rosettes are readily apparent in surface renderings of three-dimensional lattice light sheet microscopy data, but are not easily identified in two-dimensional projections (*Figure 3—figure supplements 1* and *2*), consistent with the failure of conventional light microscopy to identify these structures. Using scanning electron microscopy, we observed similar lamellar protrusions in fixed Jurkat T cells, another amoeboid cell type that employs fast, low-adhesion crawling through complex environments (*Figure 3—figure supplement 3*).

Lamellar pseudopods were not limited to the plane of the coverslip and, in three-dimensional movies, we observed many pseudopodial sheets travelling across the dorsal surface of the cell, from front to back (*Figure 2A* and *Video 1*) or side to side (*Video 2*). Importantly, most pseudopodial sheets emerged from the cell body and projected directly into the liquid medium without ever contacting the surface of the coverslip (*Videos 3* and *4* and time overlay in *Figure 2A*). Despite being unsupported, these pseudopods maintained a lamellar morphology with an approximately constant thickness throughout their lifetime (*Figure 2E*).

To verify that the distinctive lamellar shape of pseudopodial sheets does not require interaction with a flat surface, we also imaged membrane-labeled HL-60 cells crawling through random collagen networks (*Figure 2B*). Regardless of whether cells crawled across glass or through a collagen matrix, their lamellar pseudopods shared several common features, including a similar profile (*Figure 2C*) and thickness (780 ± 137 nm standard deviation, n = 18 cells). The thickness measured from surface-rendered LLSm data agrees with measurements made on other types of data, including: raw lattice light sheet data (567 ± 64 nm on glass and 689 ± 86 nm in collagen); spinning disk confocal images of phalloidin-stained actin (505 ± 49 nm for a sheet and 533 ± 123 nm for a rosette petal); and scanning electron micrographs, which suggest a thickness of ~640 nm (*Fleck et al., 2005*). Given the resolution of lattice light sheet microscopy (~230 nm in XY [*Chen et al., 2014*]), we estimate that the actual thickness of lamellar pseudopods lies between 430 and 800 nm.

Lamellar pseudopods are not perfectly flat, and exhibit varying degrees of pucker and undulation. To determine whether the flatness of lamellar sheets is influenced by cell surface interactions, we first quantified their flatness by: (i) choosing points distributed along the leading edge of each pseudopod, (ii) using them to

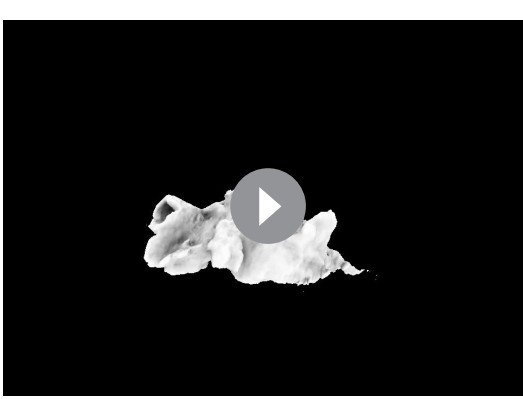

**Video 2.** Another example of lamellar pseudopods formed by cells crawling on a flat surface (fibronectin-coated glass coverslip). Video plays at 11 × real time. See also *Video 1*.
DOI: https://doi.org/10.7554/eLife.26990.007

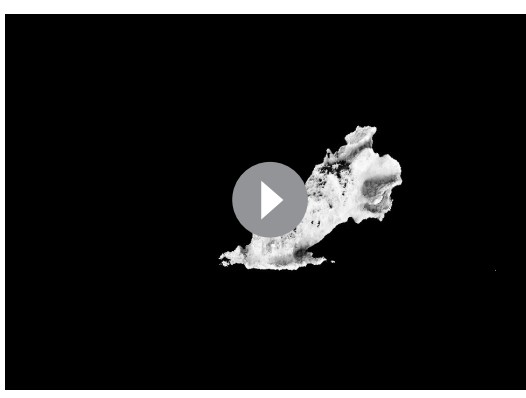

**Video 3.** Example where sheets never come into contact with surface. Because the sample is held vertically in the imaging chamber, edges of cells falling off of the coverslip can pass through the field of view, seen here as objects flowing from right to left. Video plays at 9.8 × real time.
DOI: https://doi.org/10.7554/eLife.26990.008

define a best-fit, reference plane, and (iii) calculating the average minimum Euclidean distance, or average vertical error, between the original points on the leading edge and the reference plane. Cells crawling across coverslips, as well as those crawling through collagen networks, build sheets with similar flatness (*Figure 2D*), raising the question of how cells can form flat structures without the surface template mechanism proposed for the assembly of flat lamellipodia of adherent cells (see Discussion).

## Pseudopodial sheets interleave to form complex rosettes

In addition to simple sheets, HL-60 cells also produce more complex pseudopods. When imaged by conventional widefield or confocal microscopy these protrusions appeared as irregular, three-dimensional streaks (*Figure 3—figure supplement 1*, maximum intensity projection). However, when imaged at high resolution by lattice light sheet microscopy and visualized by three-dimensional surface rendering, these apparently amorphous structures turn out to be 'rosettes,' composed of multiple, lamellar components interleaved into rose-like patterns. Rosettes are assembled by cells crawling on a two-dimensional surface (*Figure 3*, *Videos 7* and *8*), and by cells crawling through three-dimensional collagen meshes (*Figure 3*, *Video 9*, and *Video 10*). In uniform fields of chemoattractant, both lamellar and rosette pseudopods are highly dynamic and ephemeral (*Figure 4A*).

Several lines of evidence argue that the lamellar elements forming these rosettes are identical to the lamellar pseudopods described above. Firstly, both sheet and rosette pseudopods largely excluded utrophin-based actin probes *(Figure 1B)* and both types of protrusion disappeared in the presence of CK-666 (*Figure 4B*, *Video 11*, and *Video 12*), a small-molecule inhibitor of the Arp2/3 complex (*Nolen et al., 2009*), arguing that sheets and rosettes are formed by rapidly assembling, branched actin networks. Secondly, lamellae within rosettes have the same thickness (690 ± 20 nm, n = 18, *Figure 3B*) as the simple lamellar pseudopods. Thirdly, time-lapse image sequences often revealed pseudopodial sheets converging or branching to form rosettes (e.g. *Figure 4A*, *Videos 4–5*). Fourthly, in a uniform field of chemoattractant, pseudopodial sheets and rosettes both have a lifespan of approximately 60 s, from initial appearance to final disappearance (*Figure 3—figure supplement 4*). This lifetime remains similar whether cells are migrating across flat surfaces or through collagen networks (*Figure 3—figure supplement 4*), indicating that protrusion lifespan does not depend on external mechanical support. Finally, approximately 75% of both sheet and rosette pseudopods disappear in an identical manner, by collapsing into a tangle of linear structures that resemble filopodia (*Figure 3E* (rosette) and *Video 1* (sheet), see also *Figure 4A*).

## Lamellar pseudopods grow from linear ridges

The lamellar elements that form pseudopods—both sheets and rosettes—first appeared as

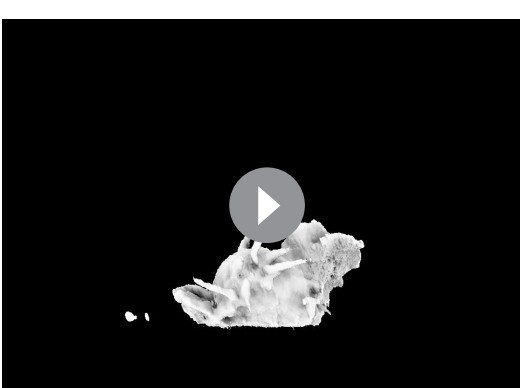

**Video 4.** Another example where sheets never come into contact with surface. Video plays at 11 × real time. See also *Video 3*.
DOI: https://doi.org/10.7554/eLife.26990.009

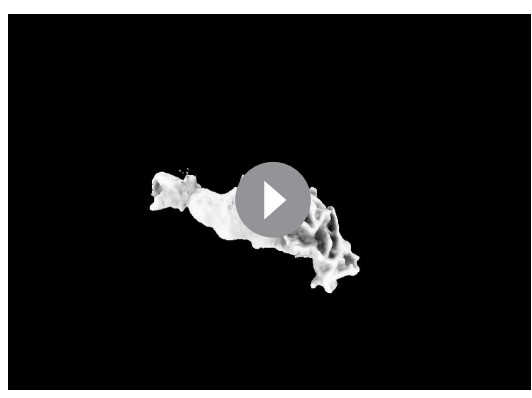

**Video 5.** Example of lamellar pseudopods formed by cells crawling through three-dimensional collagen networks. Video plays at 9.5 × real time.
DOI: https://doi.org/10.7554/eLife.26990.010

linear deformations running across the surface of the plasma membrane (*Figure 3C*). These low 'ridges' formed mature pseudopods by growing rapidly away from the cell body, at a characteristic rate of 11.9 ± 5.4 μm/min (n = 4). Remarkably, as each nascent protrusion grew, its distal edge remained linear, leaving a flat pseudopod in its wake. Isolated pseudopodial sheets sometimes produced rosettes by spawning secondary sheets, but these secondary sheets also emerged as linear ridges from the side of the original sheet (e.g. *Video 5*).

## Lamellar pseudopods in contact with the coverslip appear as dynamic 'waves' of Arp2/3 activation

Assembly of branched actin networks is largely controlled through the activation of the Arp2/3 complex, which in pseudopods is mediated by the WAVE regulatory complex (*Bear et al., 1998*; *Weiner et al., 2006*). A previous study (*Weiner et al., 2007*) employed TIRF microscopy to image localization and dynamics of the WAVE regulatory complex in HL-60 cells crawling on glass coverslips. When imaged in the region proximal (within ~100 nm) to the coverslip, the WAVE regulatory complex forms a series of thin arcs, or 'waves,' that originate somewhere in the cell body and then advance outward along the cell's ventral surface. Our observation that the pseudopod of an HL-60 cell is composed of multiple lamellar sheets suggests that each outward-moving wave is actually the edge of an individual sheet pressed into the TIRF plane as it extends forward.

To test this hypothesis, we conducted near-simultaneous TIRF microscopy and reflection interference contrast microscopy (RICM) to both visualize the WAVE complex localization and to probe the distance between the ventral surface of the cell and the glass surface on which it migrates (*Figure 5*, *Video 13*). RICM revealed that the cell's ventral surface is not smooth and does not make uniformly close contact with the fibronectin-coated coverslip to which it adheres, consistent with previous observations (*Sullivan and Mandell, 1983*). Instead, the ventral surface forms dynamic and complex patterns of close contact, including lines that travel toward the front of the crawling cell (*Figure 5*,

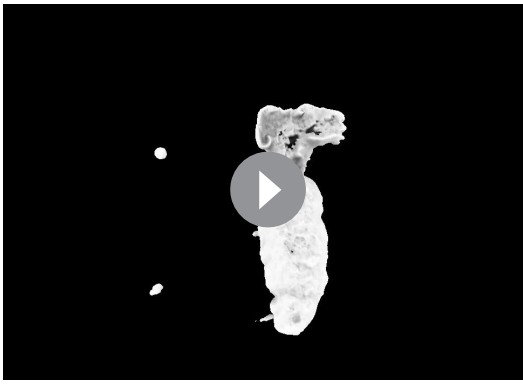

**Video 6.** Another example of lamellar pseudopods formed by cells crawling through three-dimensional collagen networks. Please note that the stage is repositioned several times because the cell migrates out of the field of view. Video plays at 10 × real time. See also *Video 5*.
DOI: https://doi.org/10.7554/eLife.26990.011

*Video 13*). Consistent with our interpretation, we also see enrichment of WAVE complex at these moving lines of close contact (*Figure 5*, *Video 13*).

## Sheet and rosette pseudopods promote changes in the direction of migration but are not required for cell locomotion

To investigate the role of dynamic pseudopods—both sheets and rosettes—in cell locomotion, we compared the morphology and motion of untreated control cells to cells treated with the Arp2/3 inhibitor CK-666 for 10 min. This analysis was limited to cells moving through three-dimensional collagen matrices because we found that the loss of Arp2/3 activity dramatically reduced adhesion of HL-60 cells to fibronectin-coated coverslips. This loss of adhesion posed particular problems for the vertically oriented coverslips used in our lattice light sheet microscope. While

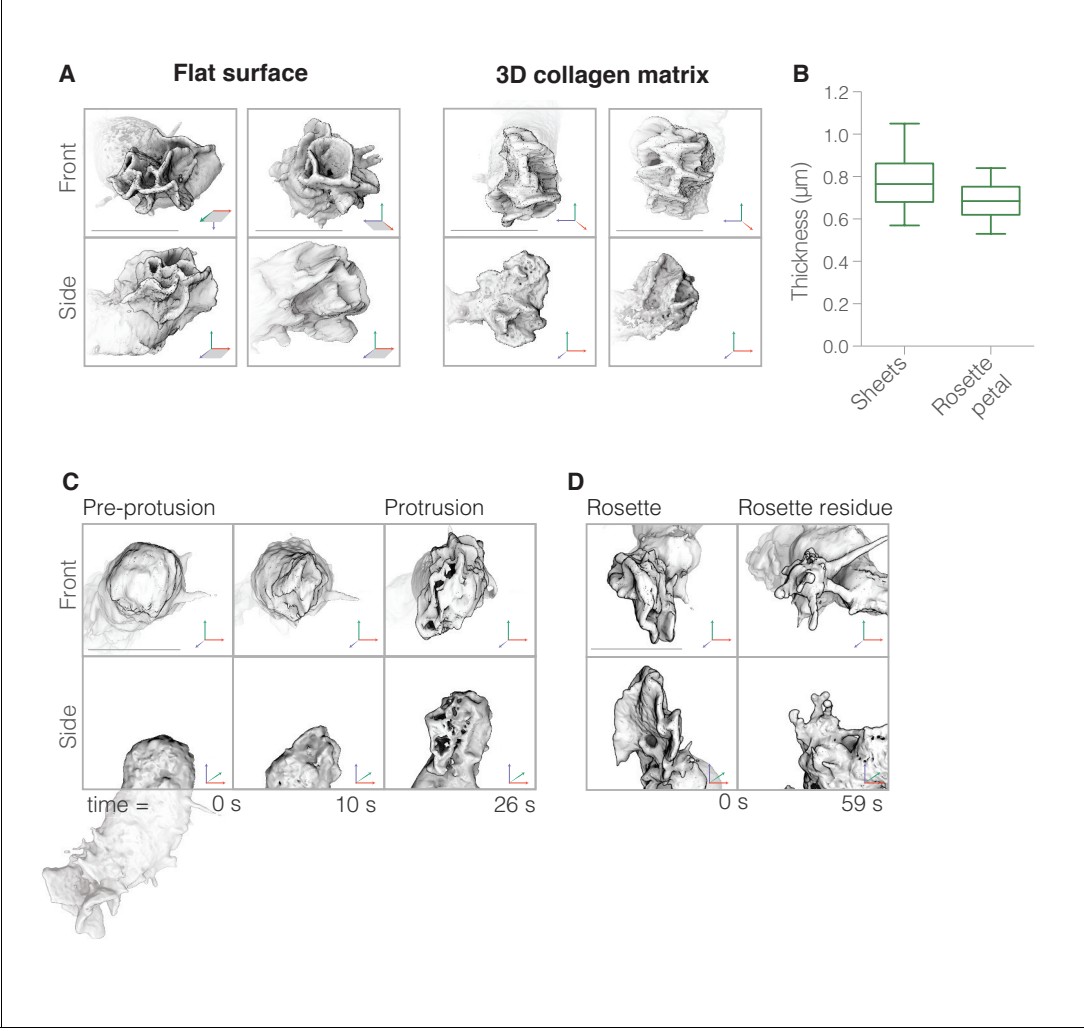

**Figure 3.** Neutrophils build complex pseudopods ('rosettes') formed of multiple lamellar sheets. (A) Three-dimensional visualizations of rosettes built by cells crawling on two-dimensional surfaces (left) as well as through unlabeled collagen meshes (right). Single timepoints of two independent cells from each condition are shown from the side (top panels) and en face (bottom panels). (B) Box and whisker plot showing that simple lamellar pseudopods have similar thickness as individual petals of complex rosette pseudopods. (n = 18 pseudopods, from 6 cells from two biological replicates). (C) Enlarged views of three time points of a portion of a single cell (whole cell shown below) showing initial stages of rosette pseudopod formation viewed en face (top) and from the side (bottom). Note that, in this instance, the new pseudopod emerges from the rear of the cell, opposite to a previously formed pseudopod. (D) Two time points showing the dissolution of a separate pseudopod shown en face (top) and from the side (bottom). For (A) and (C): Axes as indicated in *Figure 1*. Scale bars = 10 μm.
DOI: https://doi.org/10.7554/eLife.26990.018

The following figure supplements are available for figure 3:

**Figure supplement 1.** Spinning disk confocal image of fixed cell with a complex rosette pseudopod.
DOI: https://doi.org/10.7554/eLife.26990.019

**Figure supplement 2.** Maximum intensity projection of deconvolved lattice light sheet microscopy data (left) compared to surface rendering (right).
DOI: https://doi.org/10.7554/eLife.26990.020

**Figure supplement 3.** Scanning electron micrographs of Jurkat T cells on coverslips.
DOI: https://doi.org/10.7554/eLife.26990.021

**Figure supplement 4.** Box and whisker plot of lifetimes of simple (left) and rosette (right) pseudopods built by cells crawling across coated glass surfaces (blue) and through collagen meshes (brown).
DOI: https://doi.org/10.7554/eLife.26990.022

**Figure supplement 5.** UCSF Chimera visualization of two time points of published *Dictyostelium* cell dataset imaged using Bessel beam microscopy (*Gao et al., 2014*) showing rosette-like protrusions.
DOI: https://doi.org/10.7554/eLife.26990.023

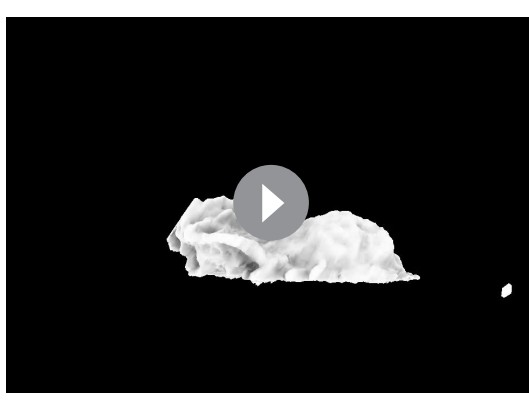

**Video 7.** Example of 'rosette' pseudopods built by cells crawling on flat surface (fibronectin-coated glass coverslips). *Video 7* plays at 11 × real time.
DOI: https://doi.org/10.7554/eLife.26990.012

control cells built lamellar pseudopods, CK-666 treated cells formed long, tubular protrusions (*Figure 4A and B*).

To analyze complex pseudopod dynamics, we developed an automated method for pseudopod detection and volume estimation in our three-dimensional datasets (*Figure 6A*). This method takes advantage of the preferential binding of the utrophin actin probe to stable structures, such as cortical actin networks, over the more dynamic, branched networks that drive membrane movement (*Belin et al., 2014*). As expected, our lattice light sheet microscopy revealed clear colocalization of the utrophin probe with the membrane marker everywhere on the cell surface *except* inside the dynamic sheets and rosettes, which largely excluded the probe (*Figure 1B*), despite containing dense actin networks (*Figure 2—figure supplement 1* and *Figure 3—figure supplement 1*). We used the absence of fluorescent utrophin from membrane-proximal regions of the cell as a convenient marker to identify protrusions driven by the growth of branched actin networks, and automated this process by subtracting the volume enclosed by a utrophin-based actin probe from the volume enclosed by the plasma membrane marker (*Figure 6A*, *Video 14*). This method of automated identification and quantification reveals that inhibition of Arp2/3 complex activity decreases the volume of the largest pseudopod produced by crawling cells by an average of 40% (*Figure 6—figure supplement 1C* and *Videos 14–15*), a significant decrease despite the within-cell variability of pseudopod volume (*Figure 6—figure supplement 2D–E*). Similarly, visual inspection of time-lapse sequences revealed that the rate of pseudopod formation decreased from 1.9 pseudopods/min to 0.8 pseudopods/min upon Arp2/3 inhibition (Control: n = 9 cells over 40.7 min; CK-666 treated: n = 7 cells over 22.1 min). The remaining pseudopods may be due to incomplete inhibition of Arp2/3.

To assess the effect of inhibiting the Arp2/3 complex on cell locomotion, we compared movement of cell centroids in the absence and presence of the Arp2/3 inhibitor CK-666. Despite the loss of dynamic pseudopods, cells treated with CK-666 retained the ability to migrate through collagen networks in a uniform field of chemoattractant. The treated cells, however, moved more slowly (5.42 vs 8.57 um/min, *Figure 6E*) and appeared to travel along much straighter trajectories than control cells (*Figure 6B*). We used two methods to quantify and analyze cell turning in three dimensions. In one approach, we detect turns by estimating the first derivative of the cell's trajectory to identify inflection points in the cell's path. The second method computes the ratio of the path length traveled by the cell centroid over ten time points to Euclidean distance between the first and last points, and defines turns as regions where this ratio is larger than 2.5 (*Figure 6C*). According to both metrics, control cells changed direction 2 to 3 times more often than the CK-666 treated cells (method 1: 2.73 vs 1.15 turns/min, *Figure 6—figure supplement 1A and B*; method 2: 3.41 vs 1.09 turns/min, *Figure 6D–F*), and displayed greater turn activity (*Figure 6—figure supplement 1B*).

The reduced turning frequency in CK-666 treated cells suggest that dynamic, Arp2/3-generated pseudopods play a role in initiating or

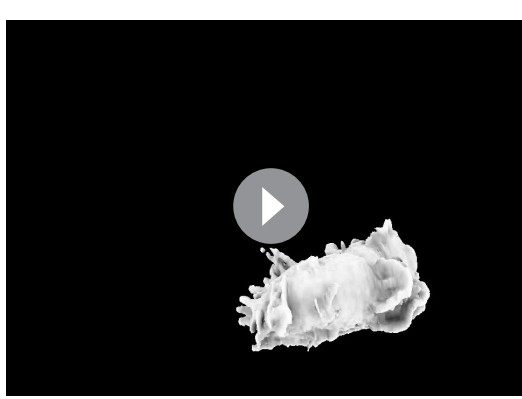

**Video 8.** Another example of 'rosette' pseudopods built by cells crawling on flat surface (fibronectin-coated glass coverslips). Video plays at 11 × real time. See also *Video 7*.
DOI: https://doi.org/10.7554/eLife.26990.013

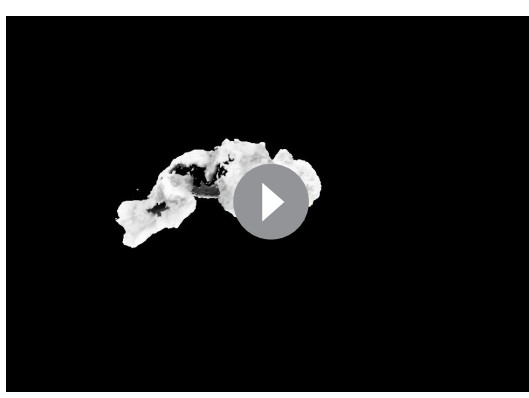

**Video 9.** Example of 'rosette' pseudopods built by cells crawling through polymerized collagen networks. Video plays at 11 × real time.
DOI: https://doi.org/10.7554/eLife.26990.014

facilitating direction changes. To further explore this possibility, we visualized the relationships between pseudopod formation and cell turning by overlaying the number and volume of dynamic pseudopods onto the three-dimensional trajectories of cells crawling through a collagen matrix (*Figure 6B*). In these plots, the line is a two-dimensional projection of the three-dimensional trajectory of the cell centroid projected to a plane that maximizes parallel coincidence, while the line width and color saturation indicate, respectively, the number and volume of dynamic pseudopods at every point. Control cell trajectories reveal frequent turns, each of which appears correlated with a dramatic increase in pseudopod dynamics. In contrast, CK-666 treated cells made less than half as many turns (*Figure 6F*) and created fewer and smaller dynamic pseudopods (*Figure 6—figure supplement 1C* and *Figure 6—figure supplement 2D–E*). Furthermore, while control cells frequently build pseudopods in the direction of cell motion, the protrusions built by CK-666 treated cells are more uniformly distributed (*Figure 6—figure supplement 2A and B*).

## Discussion

### Free-standing lamellae combine to form complex pseudopods

Lattice light sheet microscopy reveals that complex pseudopods formed by fast-moving HL-60 cells are actually composed of dynamic lamellar elements. Some pseudopods are isolated, free-standing lamellae, while others are rosettes composed of multiple lamellar elements, often connected by T- and Y-shaped junctions. These lamellar building blocks share common structural features, including a characteristic thickness, but their sheet-like morphology does not depend on interaction with a flat surface.

We identified similar lamellar and rosette protrusions in published scanning electron micrographs of several different cells types, including differentiated HL-60 cells (*Fleck et al., 2005*), Jurkat T cells (*Nicholson-Dykstra and Higgs, 2008*) (*Figure 3—figure supplement 3*), human neutrophils migrating through a chemoattractant gradient (*Zigmond and Sullivan, 1979*), dendritic cells (*Coates et al., 2003*; *Fisher et al., 2008*; *Felts et al., 2010*), T-cells (*Brown et al., 2003*) and monocytes (*Majstoravich et al., 2004*) from human blood, as well as *Dictyostelium* amoebae (*Gerisch et al., 2013*). Light microscopy has also hinted at the existence of free-standing lamellar and rosette pseudopods, notably in images of crawling cells tilted 90 degrees from the conventional viewing angle (*Sullivan and Mandell, 1983*). Using fast confocal microscopy on confined dendritic cells, the Sixt lab recently observed what appeared to be dynamic sheet-like protrusions that do not require to be adhered to any substrate (*Leithner et al., 2016*). Our rosettes also resemble protrusive structures observed in *Dictyostelium* amoebae using Bessel beam plane illumination (*Gao et al., 2012*). This similarity is heightened by surface-rendering of published *Dictyostelium* data using UCSF Chimera, and visualization techniques described

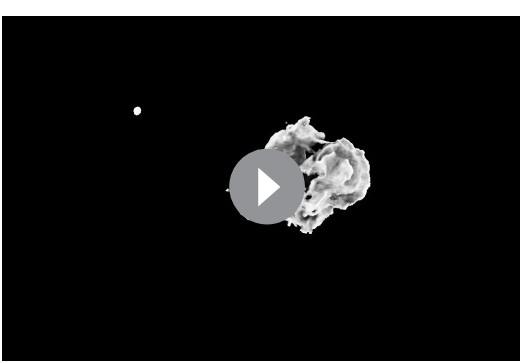

**Video 10.** Another example of 'rosette' pseudopods built by cells crawling through polymerized collagen networks. Video plays at 10 × real time. See also *Video 9*.
DOI: https://doi.org/10.7554/eLife.26990.015

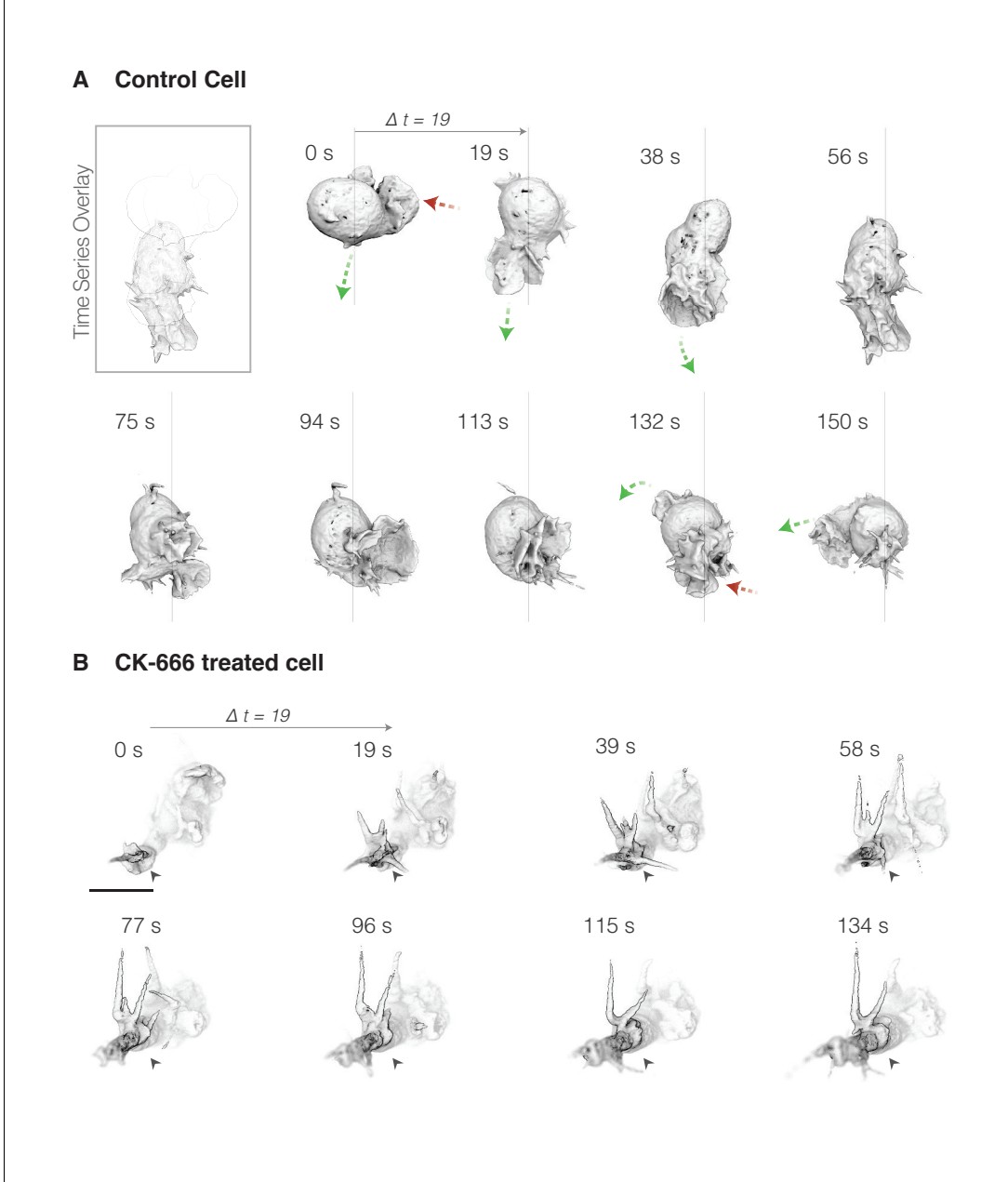

**Figure 4.** Both simple lamellar and rosette pseudopods are formed de novo from the cell body and require Arp2/3 activity for assembly. (**A**) Surface rendering of membrane-labeled control HL-60 cell at ~19 s intervals. Green arrows highlight growing pseudopods, while shrinking pseudopods are indicated by red arrows. Inset (top left): overlay of cell outline at each time point to highlight pseudopod extension and cell movement. (**B**) A similar cell treated with the Arp2/3 inhibitor CK-666 for ten minutes prior to imaging. A single plane of the three-dimensional image is darkened and indicated with a black arrow to highlight the extension of aberrant tubular protrusions. Scale bar = 10 μm.

DOI: https://doi.org/10.7554/eLife.26990.026

The following figure supplement is available for figure 4:

**Figure supplement 1.** Comparison of protrusions formed by control (Left column) and CK-666 treated (Right column) cells expressing both membrane (Top row) and cortical actin Utrophin-based actin probes (Bottom row).

DOI: https://doi.org/10.7554/eLife.26990.027

above (**Figure 3—figure supplement 5**). From our current work and previous observations, we propose two basic principles that define pseudopod morphology in these and other cells that build lamellar pseudopods. First, fast-moving leukocytes can assemble free-standing lamellar pseudopods

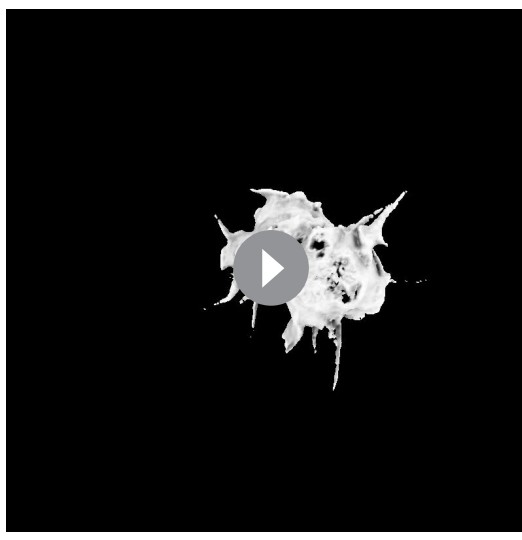

**Video 11.** Example of CK-666 treated cells migrating through a polymerized collagen network. Video plays at 11 × real time.
DOI: https://doi.org/10.7554/eLife.26990.024

without the aid of a surface 'template.' Second, more complex three-dimensional pseudopods can be resolved into lamellar components. Although suggested by previous work, establishing these facts required following the entire 'life-cycle' of membrane protrusions, from initial emergence to final retraction, using high-resolution, three-dimensional, time-lapse imaging. The lattice light sheet microscope enabled us to see the birth of lamellar pseudopods as linear ridges as well as their maturation and coalescence into more complex shapes.

What is the relationship between the free-standing, lamellar pseudopods of fast-moving leukocytes and the surface-attached lamellipodia of slower cells? Several previous studies described free-standing lamellipodial 'ruffles' peeling back from the leading edge of fibroblast or epithelial cells (*Ingram, 1969*; *Abercrombie et al., 1970*; *Abercrombie et al., 1971*), but their morphology was generally assumed to reflect transient attachment to a surface (*Burnette et al., 2014*). Interestingly, Ingram described 'sheet-like' protrusions growing on the dorsal side of adherent fibrosarcoma cells (*Ingram, 1969*), and Gauthier and co-workers discovered 'fin-like' protrusions that travel along the dorsal surface of adherent fibroblasts as they wrap around and crawl along thin nanofibers (*Guetta-Terrier et al., 2015*). These sheets and fins do not appear to form on an external substrate, but their morphology and molecular composition suggest that adherent cells may also be capable of making free-standing lamellar protrusions, independent of surface interaction. Although they resemble attached lamellipodia, the pseudopodial sheets we describe are thicker (~430–800 nm vs. <200 nm (*Abercrombie et al., 1971*)) and grow at faster rates (*Ingram, 1969*; *Abercrombie et al., 1971*). In addition, the assembly and regulation of free-standing pseudopodial sheets may require different actin regulatory molecules (*Fritz-Laylin et al., 2017*). More work is required to understand the relationships between adhesion-independent pseudopods and the lamellipodia of adhesive cells.

## Lamellar protrusions imply linear organization of regulatory molecules

How do fast-moving leukocytes produce sheet-like protrusions, even when moving through complex three-dimensional environments? Reconstitution of Arp2/3-dependent branched actin network assembly provides a simple explanation: in vitro, the morphology of a branched actin network is determined by the spatial arrangement of the regulatory molecules that activate the Arp2/3 complex. One-dimensional arrangement of Arp2/3 activator along a thin glass fiber, for example, produces a flat, sheet-like actin network (*Achard et al., 2010*; *Hu and Kuhn, 2012*), while a two-dimensional square of Arp2/3 activator produces a more three-dimensional rectangular solid (*Bieling et al., 2016*). Mathematical models

**Video 12.** Another example of CK-666 treated cells migrating through a polymerized collagen network. Video plays at 11 × real time. See also *Video 11*.
DOI: https://doi.org/10.7554/eLife.26990.025

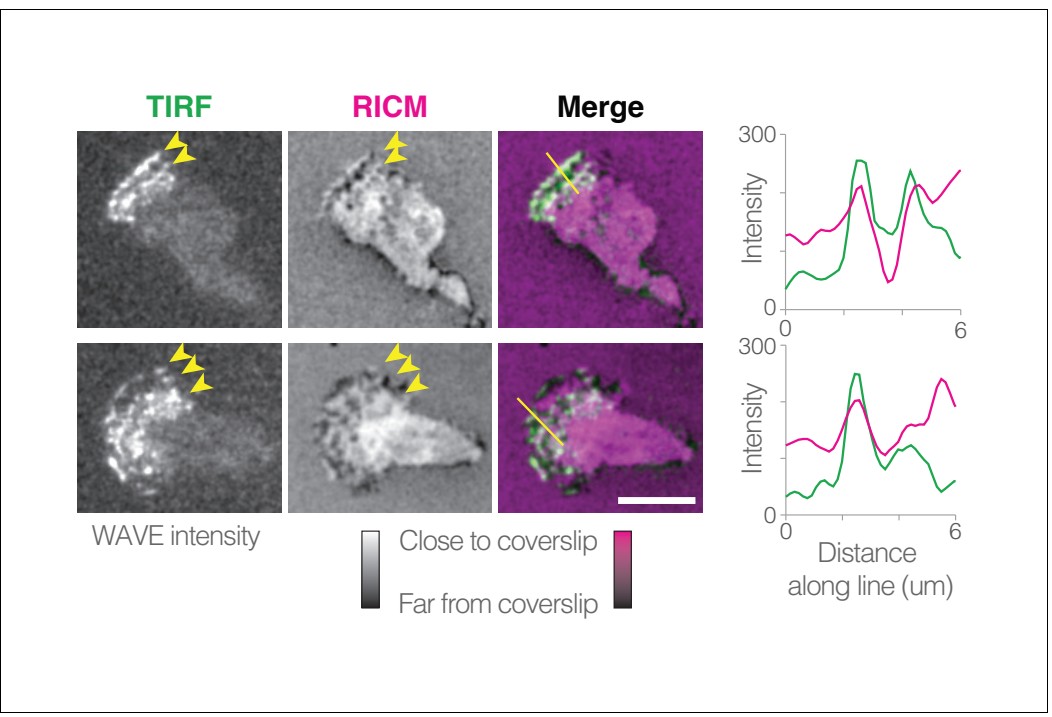

**Figure 5.** WAVE complex localizes to ventral protrusions. Time-lapse images of WAVE complex fluorescence in TIRF (left) and ventral cell surface distance from the glass coverslip in reflective interference contrast microscopy (RICM, middle), and overlay (right). For RICM, the colormap has been inverted from the raw data: dark areas represent larger distances between the coverslip and the ventral cell surface; bright white (or magenta, in the overlay) indicates close contact of the cell onto the glass. Line scans of the TIRF and RICM intensity along the lines shown on image overlays (far right). Scale bars = 5 µm. See also *Video 13* for dynamic localization and linescans.
DOI: https://doi.org/10.7554/eLife.26990.029

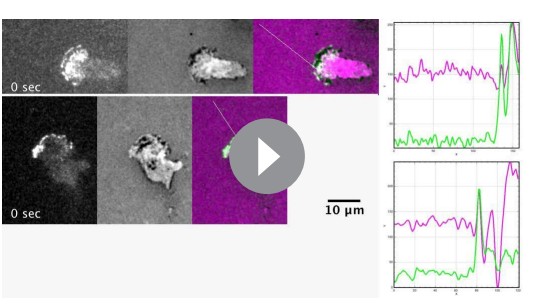

**Video 13.** Videos of WAVE fluorescence in TIRF (left) and ventral cell surface distance from the glass coverslip in RICM (middle), and overlay (right). For RICM, dark areas represent larger distances between the coverslip and the cell; bright white (or magenta, in the overlay) indicates close contact of the cell onto the glass. Two different example cells are shown. Graphs show TIRF and RICM intensity along lines drawn on image overlays (far right). Video plays at 5 × real time.
DOI: https://doi.org/10.7554/eLife.26990.028

of branched actin network formation agree with these reconstitution results and suggest that, once formed, a thin, flat actin network should remain stable as it grows (*Schmeiser and Winkler, 2015*).

Generalizing, we propose that the approximate dimensionality of nucleation and polymerase activity on the membrane is one less than the approximate dimensionality of the actin network (and protrusion) generated by this activity. In other words, a filopodium (dimension ~1) grows from a point source (dimension ~0) of nucleation/ polymerase activity while a lamellar pseudopod (dimension ~2) grows from a line source (dimension ~1). We therefore propose that the two-dimensional nature of lamellar protrusions follows as a direct consequence of the one-dimensional organization of the Arp2/3 activation machinery (*Figure 7*). We are currently exploring possible mechanisms that may give rise to this linear organization of Arp2/3 activators.

This model for the assembly of flat pseudopods is supported by previous observations that activators of Arp2/3-dependent actin assembly appear in linear patterns on the membrane. When observed at the cell surface by TIRF microscopy, direct activators of Arp2/3 activity—

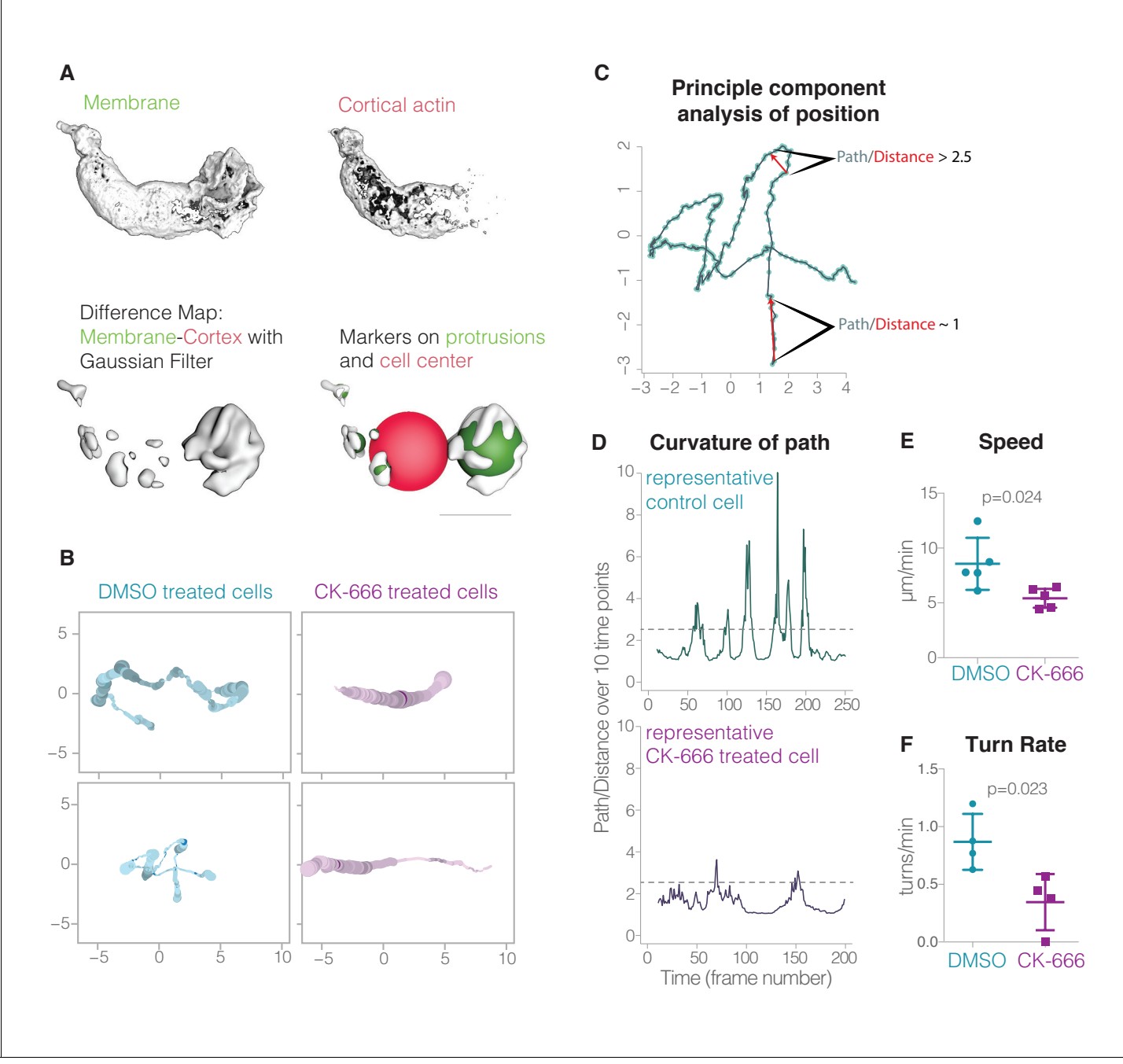

**Figure 6.** Lamellar pseudopods are associated with cell turning, but are not required for locomotion. (A) Automatic pseudopod detection relies on the relative exclusion of the Utrophin-based actin probe from the dynamic actin networks within pseudopods; pseudopods are identified by first subtracting the cortical actin signal (top right) from the membrane signal (top left), resulting in a difference map (bottom left). Pseudopods (bottom right) are then defined as enclosed volumes that fulfill all of the following three criteria: larger than 1 $\mu m^3$, within 15 $\mu m$ of cell center, and outside the cell body (or larger than 15 um$^3$). The cell body is defined by a sphere with a volume equal to the volume of the actin map; if the center of the enclosed volume was inside this sphere, it was not considered a pseudopod unless its volume exceeded 15 um$^3$. In *Videos 14,15*, the cell body is indicated by a yellow ball. (B) Plots resulting from a principal component analysis of cell position. At various points along a cell's track, the color saturation denotes the number of pseudopods and the line thickness indicates the pseudopod distribution, or the ratio of the auxiliary pseudopod volume to the total pseudopod volume. Two examples each of control cells and CK-666 treated cells are shown. (C) The results of a principal component analysis of the cell position in three-dimensional space for each time point (**blue dots**). *Low curvature*, where the path length is close to the distance traveled by the cell in ten time points, and *high curvature*, where the path length is 2.5 times longer than the distance traveled, are both indicated by red arrows. (D) Curvature of path of a representative control cell and a CK-666 treated cell. (E) Centroid speed of control cells and CK-666 treated cells (n = 5 cells from at least two independent experiments, p value = 0.024. Error bars represent standard deviation. (F) Turn rates of control cells (n = 4 cells over a

*Figure 6 continued on next page*

*Figure 6 continued*

total of 20.7 min, from at least two independent experiments) and CK-666 treated cells (n = 5 cells over a total of 13.7 min, from at least two independent experiments). Error bars represent standard deviation.

DOI: https://doi.org/10.7554/eLife.26990.032

The following source data and figure supplements are available for figure 6:

**Source code 1.** 'protrusions_chimera_script.py' Python script used to calculate protrusion volumes with UCSF chimera.

DOI: https://doi.org/10.7554/eLife.26990.035

**Source code 2.** 'cell protrusions analysis functions.R' Analysis script for use in R to calculate and plot the path characteristics, and the relationships with pseudopod activity.

DOI: https://doi.org/10.7554/eLife.26990.036

**Figure supplement 1.** Calculating turning points (A) Turn rates (calculated by two distinct methods) of control cells (n = 4 cells over a total of 20.7 min, from at least two independent experiments) and CK-666 treated cells (n = 5 cells over a total of 13.7 min, from at least two independent experiments).

DOI: https://doi.org/10.7554/eLife.26990.033

**Figure supplement 2.** Effects of Arp2/3 inhibitor on pseudopod behavior.

DOI: https://doi.org/10.7554/eLife.26990.034

WASP and the WAVE regulatory complex—form dynamic, linear arcs that propagate like traveling waves (*Weiner et al., 2007*; *Fritz-Laylin et al., 2017*). To date, only these two Arp2/3 activators have been shown to form lines, while further upstream components of the activation cascade, such as Rac, exhibit more diffuse localization patterns (*Weiner et al., 2007*; *Yang et al., 2016*).

Here, we show that these WAVE 'waves' co-localize with linear regions of pseudopods that press close to the glass surface, indicating regions of actin assembly and force generation. We propose that these multiple dynamic Arp2/3 activation zones produce multiple lamellar pseudopods that can interact to form complex rosettes. Similarly, recent lattice light sheet microscopy of *Dictyostelium* cells revealed membrane-bound circles of WAVE complex (known as SCAR in *Dictyostelium*) initiating formation of cylindrical, cup-like protrusions during macropinocytosis (*Veltman et al., 2016*). In contrast, cells that produce only a single, stable line of Arp2/3 activators at the leading edge (e.g. *Drosophila* S2 cells [*Kunda et al., 2003*] and epithelial cells [*Gautier et al., 2011*]) produce only a single lamellar protrusion.

## Lamellar pseudopods as efficient devices for searching space

We find that although complex pseudopods are not required for HL-60 cell migration, they do play a role in direction change. Using automated methods to detect and measure pseudopods we find that the size and number of these membrane protrusions increase dramatically when cells change direction and that blocking pseudopod formation reduces the frequency of direction changes (*Figure 6*). These results provide a simple explanation for recent work from the Sixt lab, who found that reducing WAVE activity in dendritic cells abolishes pseudopod formation, yet accelerates the crawling of cells in simple confined spaces (*Leithner et al., 2016*). Interestingly, the pseudopod-deficient dendritic cells fail to reorient in changing gradients of chemoattractant, and migrate more slowly toward a chemotactic source within complex three-dimensional environments compared to control cells. Similarly, in another recent paper, the Piel and Lennon-Dumenil labs found that Arp2/3 activity at the front of migrating dendritic cells was responsible for speed-fluctuating behavior (*Vargas et al., 2016*). Together with these findings in dendritic cells, our results suggest that the three-dimensional pseudopods produced by fast-moving leukocytes promote the cell turning required for exploration of the local environment.

In addition to the lamellar pseudopods of leukocytes and similar fast-moving cells, other types of protrusion advance the leading edge of other types of crawling cells, including spherical blebs

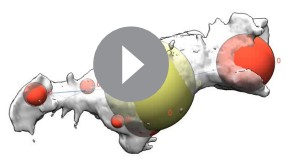

time 0

**Video 14.** Video highlighting automation of protrusion detection in control cells. Video plays at 10 × real time.

DOI: https://doi.org/10.7554/eLife.26990.030

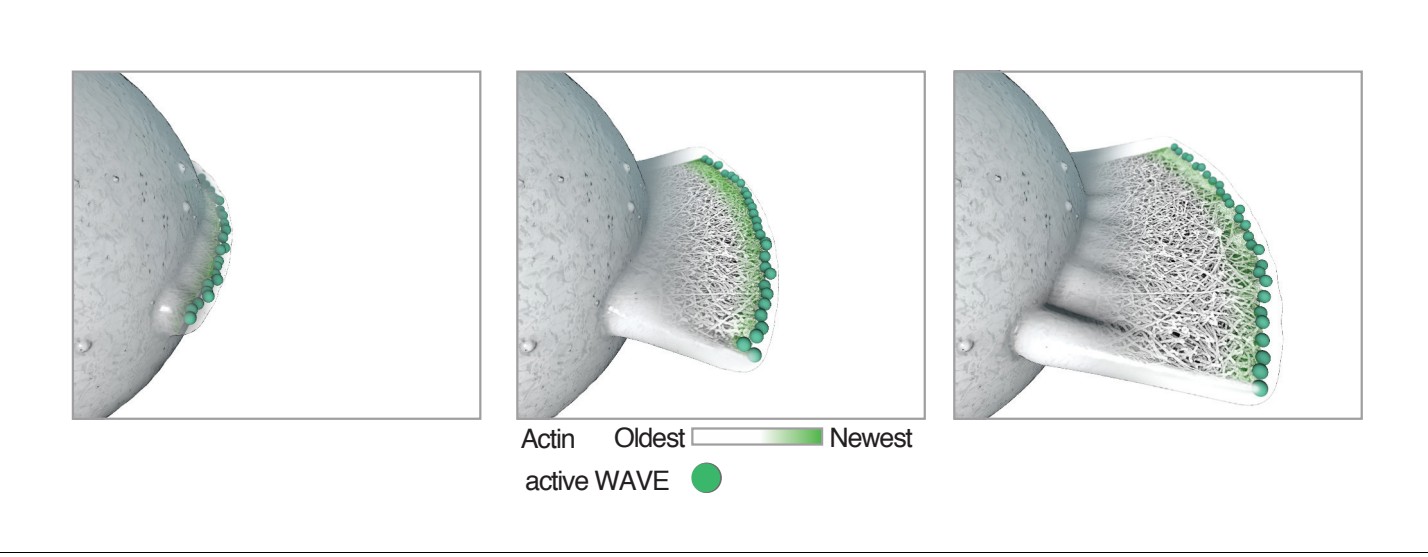

Actin  Oldest ☐▬ Newest

active WAVE ●

**Figure 7.** Model showing how localization of Arp2/3 activator (activated WAVE complex, green) could result in the formation of a lamellar protrusion independent of surface interactions. Once formed, a flat protrusion would continue growth as a planar projection as described (*Schmeiser and Winkler, 2015*).

DOI: https://doi.org/10.7554/eLife.26990.037

(*Charras and Paluch, 2008*; *Maugis et al., 2010*) and linear filopodia (*Dent et al., 2007*). There are even examples of individual cells rapidly switching between protrusion types (*Yoshida and Soldati, 2006*; *Bergert et al., 2012*; *Diz-Muñoz et al., 2016*). Do any of these morphologies provide functional, selective advantages, or do they represent an element of randomness in cellular evolution (*Bonner, 2013*)? We speculate that broad, lamellar pseudopodia and extended rosettes may be uniquely adept structures for probing a cell's environment. A pseudopodial sheet is much thinner than a spherical bleb but can sweep out a comparable volume. Linear filopodia, in contrast, would have to move both back and forth and side to side to search the same volume, and would provide limited surface contact.

## High-resolution three-dimensional microscopy requires new visualization tools

Three-dimensional time-lapse imaging with high spatial and temporal resolution poses significant challenges to the analysis tools and workflow developed for two-dimensional imaging. Specifically, lattice light sheet microscopy requires effective methods for: (i) handling large datasets (~20 GByte per time-lapse sequence); (ii) normalizing, aligning, and drift-correcting multiple fluorescence channels in three dimensions; (iii) identifying three-dimensional features of interest; and (iv) quantifying the shape and movement of complex surfaces. The most widely used open-source software for image analysis is not well suited to these tasks, and commercial software for rendering three-dimensional data is expensive (~$10,000 per license, plus yearly fees) and difficult to modify. To address these practical problems we modified UCSF Chimera to accept three-dimensional lattice light sheet microscopy datasets. UCSF Chimera is a freely available, open-source software platform developed to render and analyze three-dimensional atomic structures (*Pettersen et al., 2004*) and density maps (*Goddard et al., 2007*).

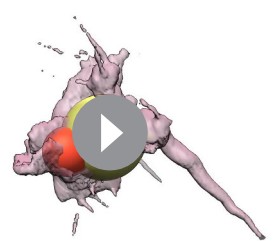

**Video 15.** *Video 15* Video highlighting automation of protrusion detection in CK666 cells. Video plays at 11 × real time.

DOI: https://doi.org/10.7554/eLife.26990.031

Most of the tools we describe here are available in the recent release and we encourage others to use them (See Materials and Methods for list of online resources).

In three-dimensional time-lapse movies, pseudopods change shape rapidly, making them difficult to identify solely by morphology. We, therefore, developed a new method for automated pseudopod detection, which exploits a biochemical peculiarity of the utrophin actin-binding domain. Utrophin localizes to the actin network that forms the cell cortex, but is largely excluded from the rapidly assembling, branched actin filaments created by the Arp2/3 complex (*Belin et al., 2014*). Our utrophin-based probe localizes to cortical actin networks proximal to the plasma membrane everywhere except within rapidly assembling networks that drive pseudopod extension. When imaged in three dimensions and combined with a marker for the plasma membrane, the absence of a utrophin-based actin probe provides a simple method to identify dynamic pseudopods and track their size and movement. Previous studies described methods to identify and track cell protrusions in two-dimensional images (*Bosgraaf and Van Haastert, 2010*), but, to our knowledge, this represents the first automated method for identifying and measuring pseudopods in three dimensions. Similar approaches will likely prove useful for defining other irregular cell structures.

Modern data visualization provides more than just pretty pictures; it can facilitate the analysis of experimental results and the formulation of scientific questions. Here, three-dimensional visualization enabled automated pseudopod detection and analysis of the relationship between pseudopod formation and cell turning. Additionally, surface rendering resolved the complex and apparently amorphous protrusions into lamellar sheets, an observation that immediately suggested a one-dimensional signaling system model. We trust that these visualization and analysis tools will provide similar breakthroughs to other areas of biological investigation.

## Materials and methods

### Cell lines

Polymerized cortical actin filament were labeled with a high-affinity actin binding domain from utrophin fused to the red fluorescent protein mCherry (*Belin et al., 2014*), and plasma membrane labeled by fusing a palmitoylation sequence from Lyn kinase (*Inoue et al., 2005*) to the green fluorescent protein mEmerald. Utrophin-mCherry HL-60 cells were derived by lentiviral transduction of cell line #CCL-240 obtained from the American Type Culture Center (ATCC), where the cell line's identity was confirmed by STR. HL-60 cell lines were grown in medium RPMI 1640 supplemented with 15% FBS, 25 mM Hepes, and 2.0 g/L NaHCO3, and at 37C with 5% CO2. HL-60 cell lines tested negative for mycoplasma by both PCR and DNA staining. Lentivirus was produced in HEK293T grown in 6-well plates and transfected with equal amounts of the lentiviral backbone vector (a protein expression vector derived from pHRSIN-CSGW (*Demaison et al., 2002*), by cloning the actin-binding domain of utrophin (*Burkel et al., 2007*) followed by a flexible linker (amino acid sequence: GDLELSRILTR) to the N-terminus of mCherry), pCMVΔ8.91 (encoding essential packaging genes) and pMD2.G (encoding VSV-G gene to pseudotype virus). After 48 hr, the supernatant from each well was removed, centrifuged at 14,000 g for 5 min to remove debris and then incubated with ~1×10^6 HL-60 cells suspended in 1 mL complete RPMI for 5–12 hr. Fresh medium was then added and the cells were recovered for 3 days to allow for target protein expression, and expressing cells were selected by fluorescence-activated cell sorting (FACS). Membrane-mEmerald and Lifeact-mCherry cell lines were derived as above and fusing the palmitoylation sequence from Lyn kinase (*Inoue et al., 2005*) to the green fluorescent protein mEmerald, and the Lifeact peptide (*Riedl et al., 2008*) to mCherry, respectively.

Prior to imaging, HL-60 cells were differentiated by treatment with 1.3% DMSO for 5 days. For two-dimensional migration, differentiated HL-60 cells were allowed to adhere to fibronectin-coated coverslips for 30 min before coverslip was moved to the imaging chamber. For three-dimensional migration, cells were overlayed onto coverslips containing pre-formed 1.7% collagen matrix polymerized from a 1:1 mixture of of unlabeled (Advanced Biomatrix catalog no. 5005) and FITC-conjugated (Sigma catalog no. C4361) bovine skin collagen, using standard protocols (*Sixt and Lämmermann, 2011*). Cells were allowed to migrate into the network for one hour before removing the coverslip to the imaging chamber. Cells were imaged in 1 × HBSS supplemented with 3% FBS,

$1 \times$ pen/strep, and 40 nM of the tripeptide formyl-MLP (to stimulation migration). Treated cells were exposed to 10 µm CK-666 (Sigma) or DMSO carrier alone (control) for ten minutes prior to imaging.

## Lattice light sheet microscopy of living cells

Time-lapse sequences generated by the lattice light sheet microscope comprised two-color image stacks, collected through entire cell volumes at 1.3 s intervals with minimal photobleaching and no evidence of phototoxicity. The spatial resolution of the resulting data was approximately 230 nm in XY and 370 nm in Z (*Chen et al., 2014*).

The imaging of HL-60 cells was carried out in lattice light sheet microscopy using Bessel beams arranged in a square lattice configuration in dithered mode, as in (*Chen et al., 2014*). For 2D migration, differentiated HL-60 cells were allowed to adhere to fibronectin-coated coverslips for 30 min before coverslip was moved to the imaging chamber at 37°C. The data was acquired on a Hamamatsu ORCA-Flash 4.0 sCMOS camera, where the moving cell was imaged by exciting each plane with a 488 nm laser at ~10 µW (at the back aperture of the excitation objective) for 5 ms, with an excitation inner/outer numerical aperture of 0.55/0.48 respectively and a corresponding light sheet length of 15 µm. At each time point, the cells were imaged by sample scanning mode and the dithered light sheet at 400 nm step size, thereby capturing a volume of ~70 µm x 90 µm x 40 µm (corresponding to $672 \times 908 \times 201$ pixels in deskewed data) every 1.2 s (which includes 1 s acquisition time and 0.2 s pause between each time point). There are 143 time points for the continuous imaging periods of 2.86 min in duration. For three-dimensional migration, mCherry- utrophin HL-60 cells were overlayed onto coverslips containing pre-formed 1.7% collagen matrix polymerized as described (14) using 50% FITC-collagen (Sigma) and allowed to migrate into the network for one hour before removing the coverslip to the imaging chamber. The cell moving inside the collagen was imaged by exciting each plane with a 488 nm laser (collagen) and 568 nm laser (HL-60) at ~20 µW and 10 µW (at the back aperture of the excitation objective) for 5 ms separately before moving to the next z plane, with an excitation inner/outer numerical aperture of 0.4/0.325 respectively and a corresponding light sheet length of 20 µm. At each time point, the cells were imagined by objective scanning mode and the dithered light sheet at 250 nm step size, thereby capturing a volume of ~70 µm x 36 µm x 35 µm (corresponding to $672 \times 352 \times 141$ pixels in raw data) every 1.4 s (which includes 1.3 s acquisition time and 0.1 s pause between each time point). There are 250 time points for the continuous imaging periods of 5.83 min in duration.

## Image alignment, compression and normalization

With the datasets we collected the imaged volume was typically five times larger than the cell along each axis so that the cell did not crawl out of view. To enable real-time playback from any vantage point we used compression and thresholding of the image data to reduce its size by as much as a factor of 100 utilizing that the cell at any instant only occupied a small portion of the imaged box. To compensate for scope drift we aligned static features (extracellular collagen filaments) of each three-dimensional image to the preceding time maximizing cross-correlation to correct microscope jitter. We normalized the intensity levels by shifting the mean to 0 and scaling the intensity for each three-dimensional image so enclosed cell volume was constant at a standard intensity value (arbitrarily chosen equal to 100). This corrected variations in normalization introduced by microscope deconvolution software and also compensated for photobleaching which reduced intensity levels by approximately a factor of 2 over the imaged time periods.

## Rendering images

The observed cell protrusions undergo continuous change in shape and can emerge from all sides of the cell. To communicate these features in two-dimensional static images we exported the surface mesh from UCSF Chimera as a Wavefront (.obj) file, and imported them into Cinema4D, a three-dimensional animation software package. We used the shading technique of ambient occlusion (recessed areas appear dark) and the Sketch and Toon Shader both with default settings to clearly visualize the three-dimensional dynamic behavior (*Figure 1—figure supplement 3*).

### Real-time viewing of the data with 'vseries' toolkit

We developed 'vseries' software within UCSF Chimera to analyze the large crawling cell data sets. The 'vseries' toolkit includes command line implementation and a Volume Series GUI. It enables users to view and manipulate an ordered sequence of volumetric datasets, and apply analysis commands to some or all of the volumetric maps at once. We used three-dimensional interactive visualization to see all aspects of the cell motion allowing any feature to be examined at any time in the cell motion. The new software visualization and analysis capabilities have been distributed as the 'vseries' command of the UCSF Chimera visualization program. To mark features of interest we also developed three-dimensional interactive surface visualizations for viewing in a web browser with the ability to hand place markers (small spheres) on desired cell features, and annotate them with text descriptions and choice of color. This software enables collaborative online annotation of the data.

### Quantification of protrusion thickness and flatness

To measure thickness of the rendered three-dimensional cells, surface meshes were imported into Cinema4D and markers were created to measure the top-to-bottom distance along the edge of a pseudopodial sheet. Flatness was measured in UCSF Chimera by first placing markers along the edge of the sheet using the 'Markers' function (http://www.rbvi.ucsf.edu/chimera/current/docs/ContributedSoftware/volumepathtracer/framevolpath.html) and then running a Python script that calculates the best fit plane and measures the distances from the plane to the markers. We also measured the thin edge of lamellar sheets intersecting raw lattice light sheet images at 90° (567 ± 64 nm on glass and 689 ± 86 nm in collagen) at half maximum fluorescence intensity). We also measured phalloidin-stained actin imaged using confocal microscopy (505 ± 49 nm for a sheet and 533 ± 123 nm for a rosette petal) intersecting the confocal plane at 90°, measured at half maximum fluorescence intensity.

### Quantification of cell movement and protrusion activity

To quantify the motions we measured cell centroid position as a function of time to obtain speed, direction, and changes of direction. This capacity is now included as the 'vseries measure' command. For each volume in the volume series, the command calculates the centroid (x,y,z) coordinates, the distance from the previous centroid ('step'), the cumulative distance along the piecewise linear path from the first centroid, the surface-enclosed volume, and the surface area. The results are saved in a text file. The centroid is the center of mass of the density map based on map regions above the threshold; we set the threshold to our already established normalized level of 100.

### Quantification of protrusion activity

To automate identification of the irregularly shaped protrusions, we took advantage of the fact that utrophin-based cortical actin labeling entered nascent protrusions slowly while membrane labeling was always present in the protrusions. We computed difference maps between the imaged membrane and actin channels, applied spatial smoothing using a Gaussian filter with a standard deviation of 0.5 μm to reduce noise. The resulting enclosed volumes were marked as individual protrusions if they were greater than 1 μm$^3$, less than 15 μm from center, and outside cell center radius or larger than 15 μm$^3$. For each volume in the volume series we calculated the total volume of all the protrusions, the distance of each protrusion from the cell center, the angle between the line connecting the cell centroid to protrusion centroid and the cell centroid path, and the number of protrusions. This calculation was implemented as a Python script (*Figure 6—source code 1*).

### Quantification of turning points

Changes of cell direction appeared correlated with emergence of multiple protrusions in different directions. To examine this correlation we used principal component analysis to find the primary plane of motion for graphing a cell centroid path in two dimensions. The first method we used for quantifying direction changes was to calculate the sliding time window ratio of the path length over the Euclidean distance between end-points using an interval of 10 time points. A ratio greater than 2.5 was considered turning. Our second approach was a Savitzky–Golay filter to calculate an estimate of the derivative. Time points were considered turnaround points if the derivative was zero

and the cell had traveled at least 1 μm from the last turnaround point. Both of these methods were written in R Studio (*Figure 6—source code 2*).

## Conventional microscopy

RICM, TIRF, and spinning disk confocal microscopy were performed on a Nikon Ti-E inverted microscope equipped with a Spectral Diskovery and an Andor iXon Ultra EMCCD. RICM illumination (also called 'interference reflection microscopy') was produced with a 50/50 beamsplitter (Chroma 21000) in the dichroic cube and a 550 nm excitation filter (Thorlabs FB550-10) after a halogen epi lamp. Sufficient neutral density filters were placed after the halogen lamp and the aperture diaphragm was adjusted to maximize the contrast of the RICM images. RICM and TIRF images were interleaved less than 500 ms apart by rotating the RICM cube in and out of the path every other frame. All hardware was controlled using Micro-Manager software (*Edelstein et al., 2010*). Image analysis of RICM, TIRF, and spinning disk confocal data was performed using ImageJ unless otherwise noted.

## Scanning electron microscopy

Jurkat T lymphocytes were obtained from the American Type Culture Collection (ATCC) who confirms cell line identity by STR, and were grown in suspension in RPMI 1640 (Gibco) supplemented with 5% fetal bovine serum (Atlanta Biologicals Inc), 2 mg/mL glucose, 1 mM sodium pyruvate (Gibco) and 50 uM beta-mercaptoethanol (Gibco), and were maintained at densities <1 million cells/mL. Cells were fixed by mixing 1 million cells with 1 vol of 7% glutaraldehyde (Electron Microscopy Sciences Inc) in 100 mM sodium phosphate pH 7.4 and incubated 1 hr at 23 degrees C. Fixed cells were washed 5x with PBS (centrifuging at 300xg for 5 min to pellet the cells), then resuspended in 100 uL PBS and allowed to settle for 20 min at 23 degrees C onto 12 mm round coverslips (Fisher) pre-coated with 0.1% poly-L-lysine (Sigma-Aldrich). Cells were re-fixed on the coverslip with 3.5% glutaraldehyde in PBS for 15 min, followed by three washes with 20 mM sodium phosphate pH 7.5, and staining with 1% osmium tetroxide (Electron Microscopy Sciences Inc) in 100 mM sodium phosphate pH 7.5. Coverslips were washed once with water, then dehydrated sequentially with 30, 50, 70, 95, and 100% ethanol. Coverslips were washed 2 × 5 min with 1:1 and 1:2 dilutions of ethanol:hexamethyldisilazane (HMDS; Sigma), then 3 × 20 min with 100% HMDS. Coverslips were dried overnight under vacuum, then adhered to SEM stubs (Electron Microscopy Sciences), coated with a 3 nm layer of osmium tetroxide using an OPC-60 osmium plasma coater (SPI Supplies), and imaged in SE mode with an FEI XL-30 FEG-ESEM using an accelerating voltage of 5 kV, spot size 3.

## Resources for getting started using Chimera for 3D light microscopy data

UCSF Chimera web site and download. https://www.cgl.ucsf.edu/chimera/

See also ChimeraX, the next generation of Chimera software, which includes vseries command. Supports ambient occlusion lighting which is not available in Chimera. http://www.cgl.ucsf.edu/chimerax/

Tutorials for general Chimera use (highly recommended for new users) www.rbvi.ucsf.edu/chimera/current/docs/UsersGuide/

Chimera Volume series graphical user interface documentation, for playing through 5d light microscopy data. https://www.cgl.ucsf.edu/chimera/docs/ContributedSoftware/volseries/volseries.html

Chimera vseries command documentation. Command to play through and process 5d image data (normalize intensities, remove jitter, extract subregions, threshold, mask, compress). https://www.cgl.ucsf.edu/chimera/docs/UsersGuide/midas/vseries.html

## Additional information

### Funding

| Funder | Grant reference number | Author |
| --- | --- | --- |
| National Institutes of Health | NIH P41-GM103311 | Thomas D Goddard<br>Thomas E Ferrin |

| Helen Hay Whitney Foundation | | Lillian K Fritz-Laylin |
| --- | --- | --- |
| National Institutes of Health | NIH/NIGMS U24 GM115370 | Megan Riel-Mehan<br>Graham T Johnson |
| Howard Hughes Medical Institute | | Lillian K Fritz-Laylin<br>Bi-Chang Chen<br>Samuel J Lord<br>Eric Betzig<br>R Dyche Mullins |
| National Institutes of Health | GM118119 | Lillian K Fritz-Laylin<br>Samuel J Lord<br>R Dyche Mullins |

The funders had no role in study design, data collection and interpretation, or the decision to submit the work for publication.

### Author contributions
Lillian K Fritz-Laylin, Conceptualization, Data curation, Formal analysis, Supervision, Funding acquisition, Validation, Investigation, Visualization, Methodology, Writing—original draft, Project administration, Writing—review and editing; Megan Riel-Mehan, Resources, Data curation, Software, Formal analysis, Investigation, Visualization, Methodology, Writing—review and editing, Figure design and illustration; Bi-Chang Chen, Data curation, Investigation, Methodology, Writing—review and editing; Samuel J Lord, Formal analysis, Validation, Investigation, Visualization, Methodology, Writing—original draft, Writing—review and editing; Thomas D Goddard, Resources, Software, Formal analysis, Investigation, Visualization, Methodology, Writing—review and editing; Thomas E Ferrin, Resources, Software, Supervision, Funding acquisition, Writing—review and editing; Susan M Nicholson-Dykstra, Henry Higgs, Acquisition of electron microscopy data; Graham T Johnson, Resources, Software, Formal analysis, Supervision, Funding acquisition, Visualization, Methodology, Writing—review and editing, Figure design and illustration director; Eric Betzig, Resources, Software, Supervision, Methodology, Writing—review and editing; R Dyche Mullins, Conceptualization, Supervision, Funding acquisition, Writing—original draft, Writing—review and editing

### Author ORCIDs
Lillian K Fritz-Laylin (iD) http://orcid.org/0000-0002-9237-9403
Samuel J Lord (iD) http://orcid.org/0000-0002-2785-989X
R Dyche Mullins (iD) http://orcid.org/0000-0002-0871-5479

### Decision letter and Author response
Decision letter https://doi.org/10.7554/eLife.26990.040
Author response https://doi.org/10.7554/eLife.26990.041

## Additional files

### Supplementary files
• Transparent reporting form
DOI: https://doi.org/10.7554/eLife.26990.038

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
