## [Decision Letter]

Thank you for submitting your article "Actin-based protrusions of migrating neutrophils are intrinsically lamellar and facilitate direction changes" for consideration by *eLife*. Your article has been reviewed by three peer reviewers, one of whom, Michael Sixt, is a member of our Board of Reviewing Editors, and the evaluation has been overseen by Anna Akhmanova as the Senior Editor. Matthieu Piel, one of the other two reviewers, has agreed to reveal his identity.

The reviewers have discussed the reviews with one another and the Reviewing Editor has drafted this decision to help you prepare a revised submission.

Summary:

In their study Fritz-Laylin et al. study the morphology of 3D migrating amoeboid cells (mainly HL60 derived neutrophils) using high resolution high speed light sheet microscopy. They find that what was previously called pseudopodial extension is actually an array of lamellipodia in varying configurations – from one single sheet to rosettes. They show this quantitatively by applying a new modality of image analysis using several reporter lines. Pharmacology is used to inhibit the Arp2/3 complex which left locomotion intact but decreased the extent of cell turning. An important insight is also that previously described waves of actin polymerisation are not an internal feature of the cell but are rather lamellipodial tips sliding along the ventral surface of the cell.

The authors conclude that lamellipodia can form independently of adhesive contacts, that lamellipodia are important for cell turning but not necessarily for locomotion and that WAVE waves are tips of expanding lamellipodia.

Essential revisions:

All three reviewers agreed that this is brilliant imaging and the conclusions are important for the community, although the study is very observational and in its functional implications overlapping with two papers in Nat Cell Biol on Dendritic cells by the groups of Lennon-Dumenil and Sixt. One reviewer feels that the work is too phenomenological for *eLife* and suggests that BAR domain proteins potentially important for lamellipodial flatness should be investigated in detail. As this will go beyond the two months revision time the reviewers agreed during the consultation process that emphasis should be put on the general morphological part and the authors should employ at least one other classical amoeboid cell type to proof the principle. T cell blasts might be a suitable and easy to handle example.

---

## [Author Response]

All three reviewers agreed that this is brilliant imaging and the conclusions are important for the community, although the study is very observational and in its functional implications overlapping with two papers in Nat Cell Biol on Dendritic cells by the groups of Lennon-Dumenil and Sixt. One reviewer feels that the work is too phenomenological for eLife and suggests that BAR domain proteins potentially important for lamellipodial flatness should be investigated in detail. As this will go beyond the two months revision time the reviewers agreed during the consultation process that emphasis should be put on the general morphological part and the authors should employ at least one other classical amoeboid cell type to proof the principle. T cell blasts might be a suitable and easy to handle example.

We have added an additional figure (Figure 3—figure supplement 3) showing Jurkat T cells also make lamellar protrusions, and have added text describing this data to the manuscript:

“Dynamic sheets and rosettes are readily apparent in surface renderings of three-dimensional lattice light sheet microscopy data are not easily identified in two-dimensional projections (Figure 3—figure supplement 1 and Figure 3—figure supplement 2), consistent with the failure of widefield light microscopy to identify these structures. Using scanning electron microscopy, we observed similar lamellar protrusions in fixed Jurkat T cells, another amoeboid cell type that employs fast, low-adhesion crawling through complex environments (Figure 3—figure supplement 3).

We have also added additional references to and discussion of published literature showing similar shaped protrusions on other cell types to the existing Discussion:

“We identified similar lamellar and rosette protrusions in published scanning electron micrographs of several different cells types, including differentiated HL-60 cells (Fleck et al., 2005), Jurkat T cells (Nicholson-Dykstra and Higgs 2008) [PMID: 18720401] (Figure 3—figure supplement 3), human neutrophils migrating through a chemoattractant gradient (Zigmond and Sullivan, 1979), dendritic cells (Coates et al., 2003; Fisher et al., 2008; Felts et al., 2010), T-cells (Brown et al., 2003) and monocytes (Majstoravich et al., 2004) from human blood, as well as Dictyostelium amoebae (Gerisch et al., 2013). Light microscopy has also hinted at the existence of free-standing lamellar and rosette pseudopods, notably in images of crawling cells tilted 90 degrees from the conventional viewing angle (Sullivan and Mandell, 1983). Using fast confocal microscopy on confined dendritic cells, the Sixt lab recently observed what appeared to be dynamic sheet-like protrusions that do not require to be adhered to any substrate (Leithner et al. 2016). Our rosettes also resemble protrusive structures observed in Dictyostelium amoebae using Bessel beam plane illumination (Gao et al., 2012). This similarity is heightened by surface-rendering of published Dictyostelium data using UCSF Chimera, and visualization techniques described above (Figure 3—figure supplement 5).”

In our Discussion of the Sixt paper, we now also included the work by Lennon-Dumenil:

“These results provide a simple explanation for recent work from the Sixt lab (Leithner et al., 2016), who found that reducing WAVE activity in dendritic cells abolishes pseudopod formation, yet accelerates the crawling of cells in simple confined spaces. Interestingly, the pseudopod-deficient dendritic cells fail to reorient in changing gradients of chemoattractant, and migrate more slowly toward a chemotactic source within complex three-dimensional environments compared to control cells. Similarly, in another recent paper, the Piel and Lennon-Dumenil labs found that Arp2/3 activity at the front of migrating dendritic cells was responsible for speed-fluctuating behavior (Vargas et al., 2016). Together with these findings in dendritic cells, our results suggest that the three-dimensional pseudopods produced by fast-moving leukocytes promote the cell turning required for exploration of the local environment.”